# Tight Margins: Compression Garment Use during Exercise and Recovery—A Systematic Review

**Alana J. Leabeater \*, Lachlan P. James and Matthew W. Driller**

Sport and Exercise Science, School of Allied Health, Human Services and Sport, La Trobe University, Melbourne, VIC 3086, Australia; l.james@latrobe.edu.au (L.P.J.); m.driller@latrobe.edu.au (M.W.D.)

\* Correspondence: a.leabeater@latrobe.edu.au

**Abstract:** *Background:* Compression garments (CGs) are a popular tool that may act on physiological, physical, neuromuscular, biomechanical, and/or perceptual domains during exercise and recovery from exercise, with varying levels of efficacy. While previous reviews have focused on the effects of CGs during running, high-intensity exercise, and exercise recovery, a comprehensive systematic review that assesses the effectiveness of garment use both during and after exercise has not been recently conducted. *Methods:* A systematic search of the literature from the earliest record until May 2022 was performed based on the PRISMA-P guidelines for systematic reviews, using the online databases PubMed, SPORTDiscus, and Google Scholar. *Results:* 160 articles with 2530 total participants were included for analysis in the systematic review, comprised of 103 'during exercise' studies, 42 'during recovery' studies, and 15 combined design studies. *Conclusions:* During exercise, CGs have a limited effect on global measures of endurance performance but may improve some sport-specific variables (e.g., countermovement jump height). Most muscle proteins/metabolites are unchanged with the use of CGs during exercise, though measures of blood lactate tend to be lowered. CGs for recovery appear to have a positive benefit on subsequent bouts of endurance (e.g., cycling time trials) and resistance exercise (e.g., isokinetic dynamometry). CGs are associated with reductions in lactate dehydrogenase during recovery and are consistently associated with decreases in perceived muscle soreness following fatiguing exercise. This review may provide a useful point of reference for practitioners and researchers interested in the effect of CGs on particular outcome variables or exercise types.

**Keywords:** sports technology; recovery; performance; compressive; endurance; strength; muscle soreness

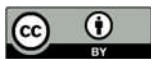

## 1. Introduction

### 1.1. Background

Sports performance is framed by the key tenets of training stimulus, fatigue, recovery, and adaptation, which, when balanced correctly, allow for supercompensation and optimal adaptation while avoiding overtraining [1]. With this in mind, sports technologies that allow athletes to improve their performance capability and/or increase their availability to train and compete optimally are highly valued [2]. Compression garments (CGs) are one such example of a sports technology that has been widely embraced by athletes across the entire competitive spectrum, from recreational to elite, as they purport to improve elements of both athletic performance and recovery [3]. Despite mixed empirical evidence for the effectiveness of CGs, their popularity amongst athletes and sports teams continues to grow; for example, it has been reported that up to 71% of elite athletes sleep in CGs at least once per week [4].

CGs are broadly defined as apparel that apply mechanical pressure to body tissues and include knee-high and above-knee socks, calf sleeves, shorts, waist-to-ankle tights,

arm sleeves, short and long-sleeve shirts, and full-body suits [5]. The relatively recent application of CGs to sporting and athletic contexts differs from their clinical and therapeutic origin, where they have been used since the late 19th century in the treatment of pulmonary embolisms, oedemas, deep vein thrombosis, and leg wounds or ulcers [6,7]. In this context, the garments were used to promote blood flow from superficial veins into deep veins, thereby preventing cutaneous venous stasis and the development of conditions such as chronic venous insufficiency [8,9]. Early research with healthy populations suggested benefits to exercise performance such as increased proprioception and reduced muscle oscillation [10], and decreased blood lactate [11] during running and cycling, respectively. Meanwhile, decreased muscle soreness [12,13], increased blood lactate removal [11], and increased perception of recovery [14] have been reported in studies focusing on the application of compression as a recovery method for athletes. However, since these early publications, there has been substantial growth in research on CGs, such that ~85% of studies in this field were published from 2010 onwards [15].

Previous systematic reviews and meta-analyses have reported few ergogenic effects for CGs during running [16] or other high-intensity exercises [17], but small positive effects on recovery kinetics after exercise [5], most likely for strength recovery [18] and perceptual elements of recovery, such as perceived muscle soreness [16]. A recent systematic scoping review by Weakley et al. [15] summarised the effect of CGs by research outcomes and variables and provided an overall assessment of research published prior to December 2020. However, as a scoping review, this did not allow for complete reporting of individual studies and categorisation into 'during exercise' and 'recovery' studies. Therefore, a comprehensive systematic review that assesses the effectiveness of garments both during and after exercise and provides full data for the included studies is warranted.

### 1.2. Objectives

The aim of this analysis was to systematically review the effects of compression garments (a) during sport and exercise performance and (b) during recovery after exercise or between exercise bouts.

## 2. Methods

### 2.1. Literature Search

A systematic search of the literature was performed based on the PRISMA-P guidelines for systematic reviews [19] using customised systematic review software (Covidence, Veritas Health Innovation, Melbourne, VIC, Australia). A search of the online databases PubMed, SPORTDiscus, and Google Scholar was performed for studies published up to May 2021 using the following keywords: compression, compression garment, compressive garment, stocking, sport, exercise, recovery, and/or performance. Relevant articles (e.g., review articles) were also used for additional reference searching. An updated search of the literature was performed in May 2022 to account for additional studies published in the period since the initial search.

### 2.2. Study Selection

Studies were eligible for inclusion if they were original research from peer-reviewed academic journals, written in English, and investigated the use of compressive garments (that permit prolonged wear) for the purpose of applying pressure to particular areas of the body to aid sport/exercise performance or recovery, using healthy participants of any training level.

Studies were excluded from the analysis if they did not yield change-score data; if a full-text report could not be retrieved; if they did not include a control or comparison group; if they were not conducted in the context of sport and exercise performance or recovery; if they utilised intermittent pneumatic compression rather than a compression

garment; and if they were conducted on clinical, injured or otherwise unhealthy populations.

### 2.3. Data Extraction and Synthesis

After the duplicates were removed and the full-text articles retrieved, an initial screening of the title and abstract was performed by two reviewers. The articles that did not meet the eligibility criteria were removed from further analysis, and any disagreements were resolved by discussion. The remaining articles were evaluated via a full-text review by the same two reviewers, with the resultant articles included for analysis.

The articles were then placed into one of three categories for data extraction: during exercise studies, recovery studies, or combined studies. The following characteristics were then extracted from each article and used to tabulate the results: publication year; study design; performance/recovery outcome(s); sample details (sample size, sex, height, weight and training history); compression garment details (type, manufacturer, composition and applied pressure, if provided); exercise modality and protocol; duration of recovery (if applicable), and effects of compression clothing. The latter are presented by statistical significance ($p < 0.05$) and/or standardised effect sizes (Cohen's *d*: *small* = 0.20–0.50, *moderate* = 0.50–0.80, *large* = > 0.80), unless otherwise stated.

## 3. Results

### Study Characteristics

The literature search identified 1175 articles, with an additional 49 articles identified through citation searching. Following the initial screening, which removed 1002 articles, 222 articles were sought for retrieval, of which a further six were removed due to the inability to access full-text copies. After the abstract and full-text screening of these articles by two reviewers, 155 articles were included for analysis in the systematic review. During data extraction, five of these articles were subsequently split into two studies due to different study designs or multiple studies in the same publication, bringing the total number of included studies to 160. These were comprised of 103 'during exercise' studies, 42 'during recovery' studies and 15 studies of a combined design (see Figure 1). The total number of participants in the included studies was 2530, of which 22% were female participants. (PRISMA Flow Diagram available in Supplementary Files).

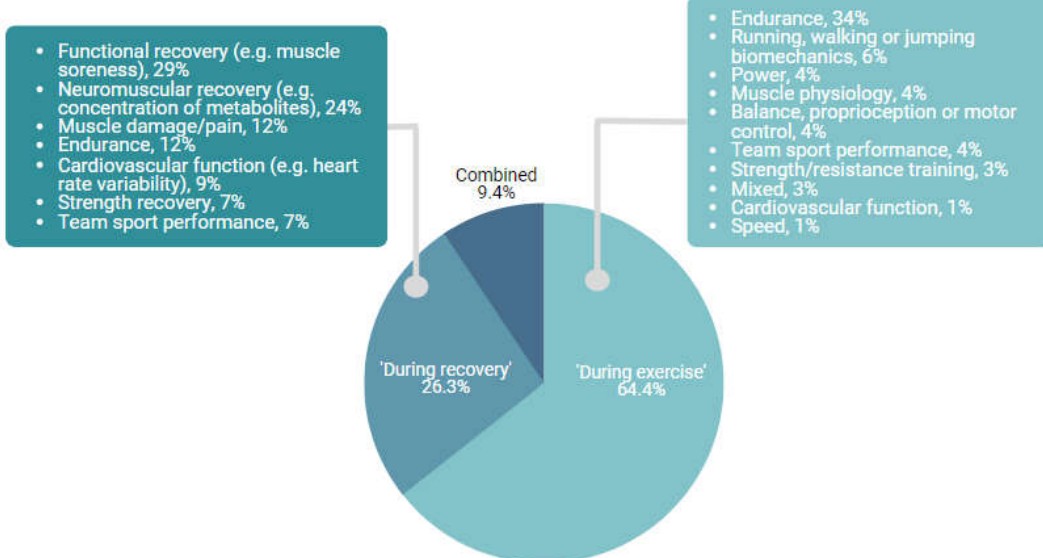

**Figure 1.** Proportion of included articles that were 'during exercise', 'during recovery' and of a combined design. Primary recovery outcome (**left**) and primary performance outcome (**right**) of the included studies is illustrated.

## 4. Discussion

Sections 4.1 and 4.2 describe the use of CGs during sport and exercise (see Table S1), while Sections 4.3–4.5 discuss outcome measures relevant to the use of CGs for sport and exercise recovery (see Table S2). Section 4.6 briefly discusses the studies that utilised CGs for both exercise performance and recovery (see Table S3).

### 4.1. Exercise Performance, by Sport/Activity

#### 4.1.1. Endurance Exercise

Given the proposed benefit of CGs on venous haemodynamics, it is unsurprising that the majority of the literature examines the influence of the garments on endurance exercise performance. The predominant modality of exercise that has been examined is running, accounting for 47% of all performance studies included in the present review. Most studies described little to no benefit of lower body CGs on measures of running performance, including finishing time in competitive marathons [20,21], ultramarathons [22], and trail runs [23,24]; distance run in a multi-stage fitness test [25]; outdoor time trials of 5 and 10 km [26,27], and measures of running economy, pace, oxygen consumption, and time to fatigue during various treadmill running protocols [28–32]. However, compression socks appear to result in a *small* improvement in time to exhaustion [33] and *small*, significant improvements to maximum speed, the speed at aerobic and anaerobic thresholds, and total work [34]. Brophy-Williams and colleagues [35] also reported that knee-high compression socks have a significant, *moderate* effect on subsequent 5 km time trial performance, though the socks utilised applied a relatively high pressure in comparison to other research (37 ± 4 mmHg at calf).

Of the 14 studies investigating cycling performance, only three examined anaerobic power using a repeat sprint protocol, showing *small* or no improvement in peak power [36–38] but a significant improvement in average power [38] with the use of waist-to-ankle tights. Cycling time trial (TT) performance with CGs has produced varying results, with a significant yet *trivial* improvement in mean power of highly trained cyclists during a 15 min TT reported by Driller and Halson [39], but no change to TT performance of 4, 6, or 10 km [40–42]. Meanwhile, Scanlan and colleagues [43] found only *small* improvements to the cycling economy and relative power in competitive cyclists during a 60-min TT. Interestingly, Smale et al. [40] reported a significant, *moderate* improvement in cognitive accuracy—as measured by performance in a cognitive Stroop task—with the application of 'low' pressure waist-to-ankle tights while cycling at 85% maximal aerobic power. However, given that there was no change to either middle cerebral artery blood flow velocity or time trial performance, the applicability of this finding to cycling performance is limited.

A small number of studies examined other modalities of endurance exercise, including walking, triathlon, and skiing, with minimal changes to the performance observed. Long-sleeved upper body compression had no significant effect on mean power output during simulated cross-country skiing [44] or lap times and velocity during a 3000 m speed skating simulation [45]; while a combination of upper and lower body CGs had a *small* positive effect (exceeding the smallest worthwhile error) on roller skiing time-trial performance in elite biathletes, though the sample size was small (*n* = 7) [46]. In a parallel group-design study, Del Coso et al. [47] employed compression socks during a competitive half-Ironman distance triathlon, finding no change to swimming or running velocity but a *small* improvement in cycling velocity, which ultimately did not improve finishing time. Meanwhile, long-sleeved upper body CGs did not influence peak or mean power, stroke rate or distance covered during a kayak ergometer maximal distance test in elite flat-water kayakers [48]. Finally, Vaile, Stefanovic, and Askew [49] reported that wheelchair rugby athletes using below-knee compression socks better maintained their average lap time during fixed-intensity wheelchair bouts, though maximal sprint time was unchanged.

Overall, these results indicate that CGs have a limited effect on global measures of endurance performance, particularly during outdoor and/or competitive settings where the garments would actually be used by athletes. Additionally, as the bulk of the literature is concerned with running and cycling performance, future research could look to observe the effect of CGs on performance measures in less common endurance modalities (e.g., rowing, swimming, hiking).

### 4.1.2. Power and Speed

Countermovement jumps have been widely used as a measure of general athletic performance or as a measure of explosive power, and in most cases, jump height (or decrement in height across a series of jumps) has improved with the use of CGs. Influenced by Berry and McMurray [50] and the use of compressive joint wraps by powerlifters, one of the earliest reported studies on the effect of lower-body CGs on performance was by Kraemer and colleagues [51] using college-level volleyball athletes. During a series of maximal countermovement jumps, the authors observed a significant improvement in average force and power production with normal-sized compression shorts in women and similar results for men in both normal and undersized conditions. More recently, Hong et al. [52] reported that compression tights significantly improved jump height and power at baseline by an average of 2.2 cm and 104 W (respectively), but there was no significant change to these measures after repeated countermovement jumps to fatigue. Similarly, significant improvements to countermovement jump height were reported by Rugg and Sternlicht [53] following a 15-min treadmill running protocol; and Doan et al. [54] following a 60 m maximal sprint. Importantly, the latter used a novel design waist-to-knee short that incorporated 75% neoprene, which may have contributed to the average 2.4 cm improvement in jump height in college-level track athletes which was reported [54]. Although, more recent studies using compression shorts (albeit with recreationally trained participants only) have not reported any significant effect on vertical jump height [55,56], drawing into question the design of the shorts in Doan et al. [54], which, due to their greater thickness and elasticity, may have contributed to the observed influence on jump height. The mechanism of this may be similar to compressive powerlifting garments (e.g., squat suits), which are used with the intention of providing additional elastic energy to enhance eccentric force [57]. The comparison between these styles of compression and their potential effect on power and velocity training could therefore be an area for future research.

When considering sprint performance, both Bernhardt and Anderson [55] and Doan et al. [54] reported no significant effect of compression shorts on 20 m or 60 m sprint time, respectively. However, Born et al. [58] reported a significant improvement in 30 m repeat sprint performance with the use of waist-to-ankle tights, although this was only for the final 10 sprints in a series of 30 and the tights themselves were of a novel design that included inner adhesive stripes of silicone around the leg muscles. Similarly, 20 m and 70 m sprint performance was not improved in elite Paralympic sprinters when using both upper and lower-body CGs, though a *small* improvement to maximal unloaded squat jump performance was observed [59]. As the sprinters had a visual impairment, the authors hypothesised that the improvement to jump performance might be related to improved proprioception, though this was not directly measured and may be of interest for further research.

### 4.1.3. Repeat Sprint and Simulated Team Sport Activities

The influence of CGs on performance has been investigated in a variety of team sports, including football (soccer), basketball, and netball. Driller et al. [60], in a study of competitive basketball players, observed no change in overall sport-specific circuit performance (the Basketball Exercise Simulation Test) but a *small*, significant improvement to 6 m sprint time measured within the circuit with the use of waist-to-ankle tights. While

there was also no improvement in countermovement jump height and power, a *small* improvement in the Margaria–Kalamen stair-climb test was noted, adding to a trend for improved lower-body power with the use of CGs [51,54,60].

In similarly designed studies, Gimenes et al. [61] and Pavin et al. [62] investigated the effect of compression stockings on measures of speed, agility, and endurance during 90-min outdoor football (soccer) matches. While the applied pressure of the stockings in both instances was particularly high (20–30 mmHg, manufacturer reported), there was no change to measures of aerobic endurance, including total distance, the distance covered at low-speed zones [61], and Yo-Yo IE2 performance [62]. However, a significant increase in distance covered at higher speed zones (>19.1 km·h$^{-1}$) and the number of accelerations [61] as well as improved T-test agility [62] were reported, all of which are important characteristics for success in an intermittent team sport such as soccer. This is a similar finding to that of Higgins, Naughton, and Burgess [63], using waist-to-ankle tights during four 15-min netball-specific circuits, who observed no change to the total distance covered but a *large* increase in the distance travelled at faster velocities (>3.5 m·s$^{-1}$). The authors proposed that CGs may improve proprioception and attenuate muscle vibrations as athletes engage in repetitive eccentric actions, which in turn may allow for the maintenance of agility and anaerobic running capacities, though without directly measuring these variables, this is purely speculative.

Finally, a combined study by Hooper et al. [64] investigated the effect of a novel upper-body compression garment on baseball and golf performance, finding a significant improvement in pitching accuracy and decrease in pitching error (baseball) and an improvement in chipping accuracy but not putting accuracy or driving distance (golf). These findings agree with previous hypotheses that CGs do not appear to improve power or endurance for complex sport skills but may positively influence coordination and accuracy by way of improved proprioception [64]. Given there is only a small number of studies that concern the performance of complex skills with CGs, greater research in this field would be required to expand this area of knowledge in a variety of sports.

### 4.1.4. Strength and Resistance Exercise

A variety of resistance exercise protocols have been used to assess the influence of CGs on strength and power outcome measures, with no significant positive effects of the garments recorded. Three similar studies of elbow flexion force using isokinetic dynamometry found no change to average torque, work, or power [65] or maximal voluntary contraction (MVC) force [66,67] with the use of upper body CGs, though interestingly, a full-sleeve upper body garment significantly improved visuomotor tracking performance 1–3 days post-exercise [66]. Similarly, bench press performance (peak and mean power, isometric strength, and muscular endurance) was unchanged with the use of arm sleeves in a single-blind study design [68]. When considering measures of lower body strength and power, muscular endurance was not improved with the use of knee compression sleeves or waist-to-knee shorts during squats to failure (>30 reps) [69,70], and MVC force was unchanged with the use of below-knee stockings during one-legged calf raise repetitions [71]. With the exception of Miyamoto et al. [71], the aforementioned investigations did not record the applied pressure of the garments used (manufacturer estimates nor the actual recorded pressure), and as such, it is difficult to ascertain whether sufficient pressure was applied to the relevant limbs to produce a noticeable effect on strength and power. It has also been suggested that with the use of machine-based resistance protocols (e.g., isokinetic dynamometers, squat machines), any potential proprioceptive benefit of CGs would be negated, as the equipment is designed to maintain the optimal movement pattern of an average-size participant [68,72], which may also play a role in the lack of observable effects of CGs in resistance exercise performance.

*4.2. By Outcomes*

4.2.1. Physiological

A total of 79 studies measured at least one physiological outcome in relation to the use of CGs during exercise, and almost half of these studies included measurement of blood concentrations of at least one muscle protein or metabolite. No changes to blood glucose [37,47,73,74], potassium or sodium concentration [21,47,75], or blood pH [36,75–77] were reported in the reviewed studies.

Of the 36 studies that measured blood lactate concentration, 25 reported no change, three reported an increase in blood lactate with the use of CGs, and seven reported a decrease in blood lactate when wearing CGs compared to a control condition. Regarding the latter, a significant decrease in blood lactate with the use of lower-body CGs was reported during or following endurance exercise protocols, including at the end of walking, and running ramped treadmill tests [78,79], respectively, and during a 10 km cycling time trial [42]. Additionally, the use of a long-sleeve upper-body CG resulted in a significant, *moderate* decrease in blood lactate during recovery from simulated cross-country skiing sprints [44]. An apparent decrease in blood lactate may reflect improved clearance or lower release from the muscle, or both [80]. Though, it is notable that performance measures (e.g., peak power, VO$_2$ max) in these protocols were not improved with CGs despite a decrease in blood lactate; and, in one instance, time to fatigue was shorter in the CG condition [79].

Well-trained runners exhibited a *small* increase in blood lactate during 6-min treadmill running bouts [33] and a 45-min submaximal treadmill run [77]; as well as a *moderate* increase in blood lactate following a treadmill time-to-exhaustion (TTE) test [33], all while using compression stockings. Similarly, in a study of elite biathletes wearing a combination of upper and lower body CGs, a *small* increase in blood lactate relative to control was reported at rest and post-TTE [46]. One explanation for this may be that local blood flow is impaired with the application of CGs, resulting in a higher muscle lactate accumulation [80]. However, given the difference in sampling methods (plasma or whole blood) and instruments used to measure blood lactate, these findings should not be considered definitive [81].

The majority of studies reported no change in measures of creatine kinase (CK, mostly measured from plasma) [21,67,73,82–84], with the exception of a *moderate* reduction in CK with the use of compression socks during a competitive half-Ironman triathlon [47]. Although, it should be noted that CK as a measure of exercise-induced muscle damage is questionable, given that it is influenced by training status, appears unspecific to the zone damaged, and is considered more accurate at establishing the occurrence of muscle damage rather than the magnitude of such damage [85,86]. An alternative—though more invasive and expensive—method to measure muscle damage is to investigate myofibrillar structure disruptions or inflammation via muscle biopsies, as was performed by Valle et al. [87]. It was found that waist-to-knee compression shorts (worn on one leg, compared to the bare leg as a control) significantly reduced histological markers of muscle damage, including intracellular albumin infiltrates and neutrophils, demonstrating an overall reduction in total muscle membrane injury following a downhill treadmill running protocol [87]. This may be a result of reduced mechanical tissue stress with the use of a CG, supporting the idea of reduced 'muscle oscillation' during intense exercise.

Mixed results were reported for the measurement of myoglobin, a protein that can be used to indicate muscle breakdown (though with the same inherent problems as the measurement of CK or blood lactate). While three reports stated no change to myoglobin [21,27,73], there were two instances of decreased myoglobin with the use of CGs: a *large* decrease following a long-course triathlon wearing compression socks [47]; and a significant decrease during the extended recovery period following a 120-min uphill treadmill run wearing hip-to-knee compression tights [74]. This may offer some practical value for athletes looking to expedite recovery following exercise; however, it is interesting that this

apparent decrease in muscle damage was not accompanied by a reduction in perceived muscle soreness or RPE during the exercise protocol [47,74]. Further, a significant *increase* in myoglobin concentration was found in recreational runners using knee-high compression stockings during a submaximal 10-km treadmill run; though without any performance or perceptual outcome measures recorded, it is unclear how this affected participants [84].

Broadly, heart rate (HR) during exercise was unchanged with the application of CGs, though eight studies reported a decrease in HR when using CGs compared to a control group. This includes significant reductions in HR during treadmill running protocols with the use of waist-to-ankle tights [31,73,76,88]; a *moderate* reduction in baseline HR when below-knee socks were applied prior to a treadmill walking protocol [78], and a *small* reduction in HR during a fixed workload cycling protocol [39]. Two studies also reported an increase in heart rate with the use of CGs; this may have been related to an increase in perfusion from the use of full-length upper-body garments, causing thermoregulatory strain that, in turn, elevated heart rate [41,48]. Though, comparatively, a *moderate* reduction in heart rate was observed when upper and lower body garments were used during a simulated sprint biathlon, without any effect on thermal comfort or strain [46]. These mixed results suggest that CGs do not reliably influence submaximal or maximal heart rate in athletic participants and, in most cases, are not correlated with performance outcomes. Additionally, no changes to blood pressure [41,84,89] or haematocrit concentration [75,82] have been reported during exercise with CGs.

Similarly, very few positive effects of CGs during exercise have been reported for other cardiorespiratory measures, including oxygen uptake and saturation, respiratory exchange ratio, ventilation, cardiac output, and carbon dioxide production. A highly cited early report by Bringard, Perrey, and Belluye [10] showed a reduction in aerobic energy cost and $VO_2$ slow component while wearing compression tights (of unspecified length and applied pressure) during indoor running, suggestive of an improvement in exercise tolerance. Given that this was conducted on a sample of only six men, more recent investigations of a similar design could be cited to demonstrate the effects of CGs on oxygen consumption at fixed workloads; for example, a significant improvement in $VO_2$ (but not $VO_2$ max) with undersized and regular-sized compression tights [31]; and, a *moderate* reduction in %$VO_2$ max attained and *large* reduction in %HR max attained during a time-to-exhaustion test while wearing graduated below-knee compression socks [33]. As both studies utilised highly-trained runners ($VO_2$ max 59–63 $mL.kg^{-1}.min^{-1}$), an improvement in the economy of oxygen use may offer some practical significance, even without concordant improvements to peak oxygen uptake and utilisation. Comparatively, measures of oxygen uptake were unchanged with the use of CGs by track and field athletes [45], recreational runners [90,91], and amateur and semi-professional team sport athletes [32,76].

Another purported benefit of wearing CGs during exercise is that they may increase blood flow to working musculature, which in turn may improve oxygen availability and exercise performance [36]. Muscle tissue oxygenation can be measured noninvasively using near-infrared spectroscopy (NIRS), though this does not distinguish between changes in blood flow to muscle tissue and changes in oxygen use, per se. Using NIRS, Coza et al. [92] demonstrated that calf compression sleeves significantly increased tissue oxygenation during the first two minutes after the start of a heel-raise exercise, most likely due to improved arterial blood flow. It was also suggested that a linear relationship exists between the relative change of total oxygenation index (TOI) and the level of externally applied compression (in mmHg), which warrants further investigation. Although, more recent studies using prolonged exercise protocols have reported conflicting results in this area, with no change to muscle tissue perfusion or maximal oxygen uptake during a trail run with calf sleeves [23] and a significant *decrease* in muscle TOI during a treadmill run with knee-high stockings, coinciding with a significant increase in intramuscular pressure in the anterior tibial muscle [84]. As discussed by Rennerfelt et al. [84], these more current

results do not suggest that healthy runners gain any circulatory benefits from wearing CGs during prolonged exercise. This is a key point of difference from early mechanistic research with CGs, which tended to show an improvement in local blood flow, but only during rest or very low-intensity activity, often using clinical populations [9,93,94].

Twelve of the reviewed studies reported at least one measure of body or skin temperature, which is particularly relevant in three of the studies that were conducted in hot environmental conditions. Goh et al. [95] reported a positive *moderate* effect of waist-to-ankle tights on time to exhaustion while running at $VO_2$ max velocity at 32 °C. However, in a single-blind research design using a 'sham' (oversized) CG, Barwood et al. [96] found no effect of a lower-body garment on 5 km time trial performance or running pace profile at 35.2 °C, despite the garment applying higher pressure at both the calf and thigh than Goh et al. [95]. In both instances, core body temperature (aural or rectal) was not influenced by the garment condition; however, a significant increase in temperature at the quadriceps was reported for both the sham and CG conditions in Barwood et al. [96], though thermal comfort and sensation was unaltered. Similar results were observed during cycling at 32 °C, where performance, as well as thermal and sweating sensation, remained unchanged (no measure of skin/body temperature was provided), despite the CG in question being a whole-body garment [97].

Paradoxically, a handful of studies found an increase in skin temperature with the use of CGs in temperate conditions, even though the garments in question did not have significant body coverage, e.g., ankle-high socks, shorts, and short sleeve garments [54,98,99]. In one instance, the use of a short-sleeve 'heat dissipating' CG (designed to improve sweat evaporation) *increased* thermoregulatory strain in trained older adults, as indicated by a significant increase in core and body temperature during submaximal cycling, coinciding with a significant increase in thermal sensation [100]. A comparable study in untrained adults found the same garment significantly reduced body temperature during passive recovery from a similar cycling protocol [75] but otherwise did not influence performance or physiological measures during exercise. The inconsistency between these results may be related to the training status of participants-given that trained athletes tend to have an earlier onset of sweating and greater sweat volume [101], it is possible that the upper body garment created a humid environment within the clothing shell, thereby increasing heat storage and thermoregulatory strain in the trained athletes. Indeed, a significant increase in covered skin temperature, forearm perfusion and vapour pressures at the chest and scapula was reported by MacRae et al. [41] with the use of a full-length upper body garment during cycling in temperate conditions. Coincidingly, the whole-body sweat rate was greater, and the comfort ratings of the CGs were lower when compared to the control condition [41]. It is therefore worth considering that upper-body CGs may act as a barrier to heat transfer in trained athletes even in temperate conditions, and the risk of adverse thermal effects–without any significant performance benefits–should be taken into account before they are used.

### 4.2.2. Biomechanical

When reviewing the 30 studies that reported the effect of CGs on biomechanical outcome measures, there was a focus on muscle activation and gait/stride mechanics during running and jumping activities. Step length and frequency were unchanged with the use of compression socks and calf sleeves during treadmill running and outdoor trail running, respectively [23,29]. In contrast, waist-to-ankle tights (of a novel design, with inner adhesive stripes around musculature) significantly increased step length but not step frequency during repeated 30-metre sprints [45]. As well, greater activation of the rectus femoris and a decreased hip flexion angle were observed, which may have contributed to the significant improvement in repeat sprint performance for the final 10 sprints of the protocol [45]. Reduced hip flexion was also observed during a 60-metre sprint with novel compression shorts in Doan et al. [54]. Other studies have reported a significant reduction in muscle activation (via EMG) with the use of CGs while running, including reduced

gastrocnemii activation during submaximal treadmill runs using either waist-to-ankle tights or below-knee stockings [102,103]. This may indicate greater muscle efficiency and, in turn, a reduction in muscle fatigue, as proposed by Bajelani, Arshi, and Akhavan [104], who demonstrated a reduction in the complexity of behaviour when participants wore compression shorts during submaximal treadmill running. However, without any observation of performance measures (and only limited physiological measures), the practical application of these findings as they relate to running performance is limited.

Biomechanical outcomes have also been investigated in relation to injury prevention with the use of CGs. For example, Chaudhari et al. [105] demonstrated a significant reduction in adductor longus activation during all five phases of stance when participants completed run-to-cut manoeuvres wearing 'directional compression shorts', which had additional bands of compression across the hip adductor muscles. While reduced muscle activation is not always correlated with reduced muscle demand, the garments may be of use in a rehabilitation setting when athletes are returning to more complex cutting or pivoting movements after an adductor injury [105]. A similar design of shorts was implemented by Bernhardt and Anderson [55] during a comprehensive sports test battery, where only the active range of motion during hip flexion was improved with the use of the shorts. CGs as prophylactic braces, therefore, do not appear to improve performance but may offer mechanical support in a similar manner to athletic/kinesiology tape [55]. Meanwhile, De Britto et al. [56] reported *moderate* reductions in knee valgus at initial contact and landing during forward, drop and vertical jumps when wearing waist-to-knee shorts, which suggests improved joint control during jump landing. However, it is difficult to extrapolate how this may relate to ACL injury prevention—as suggested by the authors—given that the garment did not cover the knees, and similar research has not been conducted with full-length compression tights. Given that a large proportion of athletes use CGs as an injury prevention tool [3], future prospective research that considers the incidence of injury among CG users may be valuable to improve what is known in this area.

One potential effect of compression garments on performance, which is often cited but rarely mechanistically investigated, is the effect of the garments on 'muscle oscillation', which more correctly should be termed soft tissue movement. The movement and vibration of soft tissues during dynamic movement is a natural protective function, which serves to dampen impact forces [106]. However, over time this repeated exposure to vibrations can lead to detrimental effects such as reduced muscle contraction force and the development of lower-body injuries [106], which has led to research on how CGs may attenuate such vibrations and soft tissue movement. The most thorough study in this area is Broatch et al. [28], who conducted a two-pronged investigation which found a significant reduction in calf and thigh muscle displacement at various speeds of treadmill running, using three different types of commercially available waist-to-ankle tights (compared to no tights). Additionally, soft tissue vibrations in the gastrocnemii and vastii muscles were significantly reduced (with 2XU brand tights only) [28]. While these are promising findings that suggest CGs may attenuate muscle displacement during repeated contractions (such as that required for running), there was no effect on running economy; however, this may be partly due to the short length of the treadmill bouts used (3 min per running speed) and the use of a treadmill as opposed to overground running, both of which would limit the ability to observe a true effect of the garments on running economy. Similarly, Dandrieux, Thouze, and Rossi [107] demonstrated a reduction in peak acceleration at varied speeds and slopes during treadmill running when participants wore waist-to-knee compression shorts; however, this was speculatively linked to the idea of 'muscle tuning' without any further interpretation. Reduced 'muscle oscillation' upon landing from a maximal vertical jump (determined from maximum longitudinal and anterior-posterior displacement of the thigh relative to the hip and knee, in cm) was also reported by Doan et al. [54], though the method used (one 60 Hz video camera) would now be considered quite dated.

Only a small number of studies have considered biomechanical outcomes in relation to modalities of exercise other than running and jumping. The effect of the garments on balance has been inconsistent, with no changes to timed balance test performance [55] and Y balance test performance (when compared to kinesiology tape applied on the thigh) [69], but improvements to single-leg balance during a specialised Biodex protocol (though in this instance, details of the CGs used were sparse) [108]. In a study of elite alpine skiers, a combination of compression shorts and socks resulted in a deeper tuck position and reduced vibration at the thigh than a normal skiing race suit, which hypothetically could reduce drag [109]. Compression tights do not appear to influence vertical ground reaction force, stiffness or height displacement during single-leg hopping to volitional exhaustion when compared to 'sham' athletic taping of the knee and ankle [110]. However, groin-to-ankle compression leg sleeves had a *small* positive effect on countermovement jump height and ground contact time of elite handball players during a handball-specific circuit; but this was not associated with any significant improvements in handball performance [111]. Interestingly, compression shorts (with a relatively low level of applied pressure: 6–15 mmHg) significantly reduced vibration transmissibility at the thigh during cycling when participants were subject to externally imposed vibration designed to mimic that experienced during regular outdoor cycling [112]. Whether this actually reduced energy demand and improved cycling performance—as suggested by the authors—is unclear, given that only heart rate and RPE were measured, and no changes were reported [112].

### 4.2.3. Perceptual

Of the 59 studies that measured perceptual outcomes, 35 recorded no change; 17 reported a decrease in the rating of perceived exertion (RPE), muscle soreness or limb fatigue (of statistical significance or greater than a *small* effect size), and the remaining seven studies reported an increase in measures of pain, soreness, discomfort or RPE (of statistical significance or greater than a *small* effect size), with the use of CGs during exercise.

Consistent with previous reviews of CGs during exercise, there was a trend toward improved perceptions of muscle soreness, RPE and/or fatigue with the use of CGs [5,16]. It has previously been suggested that as CGs apply external mechanical pressure to working muscles, they may attenuate excessive soft tissue displacement and possibly prevent structural damage, in turn improving perceived comfort and reducing fatigue or soreness [16,54,87]. However, it may also be that athletes simply perceive compression tights as more comfortable than regular athletic clothing [53], and this accounts for the improvements in perceptual responses that have been reported.

With the exception of Sperlich et al. [44], who investigated the use of a long-sleeved upper body CG during simulated cross-country skiing, the studies that reported a negative effect of CGs on perceptual responses all used below-knee stockings/socks with greater than 18 mmHg of applied pressure at the ankle (though these were predominantly manufacturer reported values). Indeed, in two studies that reported a significant increase in perceived pain and tightness and a decrease in perceived comfort, applied pressure readings of 26–32 mmHg at the ankle were measured for the 'high' level compression stockings used [27,83]. These studies reported few positive effects of CGs on performance or physiological measures, suggesting that athlete comfort is somewhat linked to the efficacy of the garments. This is highlighted in Ali et al. [27], where pre-post countermovement jump height was best maintained in 'low' and 'moderate' level compression stockings but not in 'high' level stockings, which were clearly uncomfortable for participants based on perceptual responses. By design, compression stockings/socks are able to exert a higher level of pressure than other lower-body CGs, and as such, care should be taken that these garments are perceived as comfortable by athletes prior to their use during exercise.

The potential placebo effect from the use of compression garments has been a longstanding point of discussion, given that it is difficult to control for or eliminate this effect in study design. Only two studies reported a double-blind research design [27,83],

both of which used differing grades of compression stockings with the same design that were, ultimately, correctly identified by participants 'based on feelings of perceived tightness' [27] (p. 1391). A further four studies used a single-blind design where identical compressive and non-compressive garments were employed; though, again, the participants in Gimenes et al. [61] were able to '[answer] correctly about the version of socks they used' (p. 2013). These reports highlight the inherent problem in single or double-blind research design in this field, which is that participants are clearly aware when they are wearing a compressive garment, and therefore the design itself may be inappropriate. Valle et al. [87] and Webb and Williams [113] offered a novel alternative to the problem by designing shorts and tights (respectively) that only covered one leg of the participant, allowing the other leg to act as a control; though whether this is indeed comfortable for participants is unknown, as the garments used were not intentionally designed to be one-legged.

Another way to account for the placebo effect in an unblinded research design is to record participants' *a priori* beliefs about compression garments, as in Stickford et al. [29], where positive or negative beliefs about CGs were seen to influence biomechanical and metabolic responses to submaximal running. However, while placebo effects have historically been seen as a nuisance that needs to be controlled for in research, emerging neurobiological theory suggests that researchers and practitioners can leverage this effect to improve athletic performance [114,115]. As such, the so-called placebo effect may often result in a meaningful change in performance for an individual athlete [116], and as sport becomes increasingly competitive and monetized, these small but meaningful changes should be valued.

### 4.3. Recovery of Performance Capabilities

4.3.1. Endurance Exercise

While 18 studies used endurance exercise protocols prior to recovery with CGs, only six of these studies specifically reported outcome measures relevant to endurance exercise performance, such as time trial (TT) or time-to-exhaustion (TTE) performance in running or cycling. Armstrong et al. [117] demonstrated a significant improvement in treadmill running TTE after a 48-h recovery period with below-knee socks following a competitive marathon. It is noteworthy that this was also designed as a double-blind study using 'minimally compressive' placebo socks; however, given that only manufacturer-recorded pressure readings were provided for the experimental socks, it is not known whether the placebo socks were indeed non-compressive. Meanwhile, Brophy-Williams et al. [118] used compression socks for a 60-min recovery period between maximal treadmill running bouts, showing a *moderate* effect on performance decrement in the compression group compared to the control. In both instances, the participant groups were well-trained runners, suggesting there may be a performance benefit to wearing compression socks between maximal running bouts that is worthy of consideration.

Similarly, there were generally positive results reported for measures of cycling performance, including a significant improvement in average power and TT result for a simulated 40 km event [119] and significant improvements in measures of maximal power output during 5-min bouts [11,120]. With the exception of Chatard et al. [11], who used compression stockings (and had a significantly older sample population: 63 ± 3 years old), these investigations all demonstrate the performance benefit of compression tights worn by well-trained athletes for a minimum 60-min recovery period; for example, a 1.2% improvement in 40 km TT finish time [119]. The potential for these small but worthwhile improvements in cycling performance with the use of CGs may therefore be relevant for cyclists competing in multiple events on the same day (e.g., track cyclists).

4.3.2. Sprinting, Jumping and Simulated Team Sport Activities

Only six studies investigated the recovery of sprinting, jumping and team-sport-specific capabilities after recovery with CGs, with the recovery time varying from 9.5 h to 48

h. Interestingly, three of these studies implemented a placebo recovery intervention rather than a control group, including a 5-min sham ultrasound treatment on the lower legs [121], non-compressive placebo tights [122], and a sham 'recovery drink' [123]. While intended to mitigate the potential placebo effect from the use of CGs, the lack of an actual control group (i.e., no recovery intervention) in these study designs means it is difficult to rule out the possibility of a placebo effect from the sham recovery intervention as well, and therefore the 'true' effect of the CGs is not known. Nonetheless, these investigations did not demonstrate any effect of the recovery interventions on countermovement jump performance [121,123], but there were *small* to *large* positive effects on sprinting performance [121,122]. This may be particularly relevant to rugby union players, given that the chosen exercise modality in two of these investigations was simulated rugby match-play/circuits using well-trained male rugby union athletes [122,123].

The two remaining studies investigated outcome measures relevant to basketball and volleyball performance, finding no change in vertical jump height or 5-0-5 agility performance but a *small* decrease in 20 m sprint time in competitive basketball players after wearing compression tights for 15 h [124]; and a *moderate* improvement in countermovement jump height and mean velocity in elite volleyball players after wearing compression socks for ~9.5 h [125]. In the latter, CGs were implemented by players during a long-haul flight to an international competition, showing a practical application of the garments and the potential for them to minimise some of the detrimental effects of long-distance air travel, e.g., oedema in the lower legs [125].

As discussed previously, countermovement jumps are regularly used as a measure of general athletic performance, and in this case, as a measure of functional recovery following fatiguing exercise. The majority of reports did not find any significant change to CMJ height or force when CGs were used for recovery after repeated sprint protocols [121,126], resistance exercise protocols [127,128], a cross-country skiing competition [129], or a simulated rugby union match-play [123]. However, it appears that CGs (in particular, waist-to-ankle tights) may improve functional recovery in recreationally active individuals after fatiguing drop jump protocols, as indicated by a smaller decrement in CMJ height compared to control groups [130,131]. The minimum time period for which CGs were worn in these studies was 12 h, with improvements in CMJ height reported at 24, 42 and 72 h following the exercise session [131]. Additionally, Hill et al. [130] demonstrated that CMJ performance improved after 72 h wearing 'high' pressure tights compared to 'low' pressure tights ('high': 24.3 ± 3.7 mm Hg at the calf; 'low': 14.8 ± 2.1 mm Hg at the calf). While the authors postulated that this might be due to the enhanced repair of muscle contractile elements with the greater applied pressure, it is equally likely that a placebo effect was at play, given there was no standalone control group used, only a 'sham' ultrasound for comparison [130].

### 4.3.3. Resistance Exercise

An overall positive response to CGs was reported in the seven studies that measured outcome measures related to strength recovery following various resistance exercise protocols. In early investigations with both male and female subjects, Kraemer et al. [12,13] showed a significant reduction in the decrement in peak torque and power with the use of compression arm sleeves for 5 days straight following a fatiguing eccentric arm curl protocol. More recently, the recuperation of maximal voluntary isometric contraction (MVIC) force (knee extension) was significantly improved with the use of upper- and lower-body CGs for 8 h after a high-volume protocol of maximal isokinetic eccentric and concentric knee extensor contractions [132]. When looking at recovery from hypertrophy-focused protocols, Goto and Morishima [133] reported a significant improvement in the recuperation of MVIC force (knee extension) when a whole-body CG was worn for ~18 h following a whole-body workout (in addition to anaerobic cycling sprints and drop jumps). Similarly, bench press throw power and recovery of 1RM chest press were significantly improved with the use of a whole-body CG for 24 h following a full-body workout

by resistance-trained participants [127]. The only study to use a protocol where participants performed reps to volitional failure was Hettchen et al. [134], who then applied compression tights for ~24 h, resulting in a significant improvement in maximum isokinetic hip and leg extensor strength. The variety in resistance exercise protocols used in this research broadly demonstrates that CGs may be of potential benefit for functional and neuromuscular recovery following isokinetic dynamometry and hypertrophy-focused workouts, compared to no recovery intervention at all. However, there is scope for further research investigating the effects of CGs on recovery following heavy strength and/or power training, as these methods exert a different stimulus on the neuromuscular and metabolic systems and, therefore, may result in a different recovery timeline for the participant/athlete [135].

### 4.4. Physiological Effects during Recovery

A variety of measures have been used to ascertain the influence of CGs on recovery at a physiological level, predominantly concerned with blood concentrations of muscle proteins or metabolites. There appears to be a trend for reduced blood/plasma concentrations of lactate dehydrogenase, with significant reductions following 60–80 min of recovery with CGs after submaximal [120] and maximal [11] cycling, as well as upper body [136] and lower body [137] resistance exercise protocols, compared to control groups. As well, whole-body garments were associated with a significant reduction in blood lactate following a whole-body resistance workout [127] and a *small* reduction in blood lactate 24 h following a competitive trail race [138]. However, these studies should be considered in light of the seven reports that showed no change in blood lactate [13,118,119,133,139–141] compared to control groups, once again underlining the inherent difficulty in making definitive claims about the effect of CGs based on a variable blood measurement.

Similarly, the majority of investigations did not find any change in measures of creatine kinase [12,13,128,131,133,139,141–144], though clearly, there was substantial variation in the type of exercise protocols used, which makes it difficult to establish the relevance of CK as an index of muscle damage in the first instance. With this in mind, the studies that reported significant reductions in CK with the use of CGs tended to implement explosive, eccentric-focused exercise modalities, e.g., repeat sprint and drop jump protocols [121,123,127,130,134,145] and heavy resistance exercise [137], which are known to elicit greater levels of muscle damage than endurance and non-load bearing exercise [146]. Inflammatory cytokines such as plasma IL-6, tumour necrosis factor alpha, and C-reactive protein are also commonly measured to establish the influence of CGs on systemic inflammation and tissue damage; however, studies have generally not reported any significant effect of the garments on these measures [125,128,129,133,134,137–141,144]. Though clearly there is value in measuring objective markers of muscle damage and/or inflammation to establish the effect of CGs on recovery, it is also important to recognise and acknowledge the limitations of such markers when conducting this research.

Another method to ascertain the effect of CGs on muscle damage is to observe T2-weighted magnetic resonance images, which can be used to assess the accumulation of intramuscular fluid and therefore quantify exercise-induced muscle damage [147]. However, the only study to use this method did not find any changes to T2-weighted signal intensity and intramuscular oedema (in the medial gastrocnemius) following 60 h of recovery wearing a compression sock on one leg only, after weighted heel raises to volitional fatigue [142]. This within-subject, crossover design overcomes some of the apparent 'placebo effect' of wearing CGs while also addressing inter-individual differences in DOMS and CK expression; however, the authors speculated that a systemic healing response from the body might influence the recovery of both calves regardless [142]. Similarly, Trenell and colleagues [148] used phosphorous magnetic resonance spectroscopy to assess muscle damage after participants wore compression tights for recovery on one leg only for 48 h. The results showed a significant increase in phosphodiester in the compression leg one hour following a downhill walking protocol, which may reflect increased skeletal

muscle membrane turnover and thereby accelerated muscle repair with the use of compression [148]; though this remains the only study to investigate these biochemical effects following eccentric exercise.

Some recovery studies also evaluated limb swelling following fatiguing exercise as a symptom of exercise-induced muscle damage [149] that may be mitigated with the use of CGs. It appears that CGs tend to have a beneficial effect, with significant reductions in calf girth and cross-sectional area [118,125], and upper arm and thigh circumferences [133], as well as *small* reductions in forearm circumference and midthigh girth [150], reported following a minimum 60 min of recovery with the relevant type of CGs (i.e., upper body CGs correlate to reductions in forearm circumference). Given that CGs have been used for many years in clinical settings to reduce oedemas, a reduction in limb swelling is a reliable benefit of the garments, assuming they apply a high enough pressure to exert an effect on body tissues. Future research should consider, however, that the time course of peak muscle swelling can be up to 5 days post-exercise (dependent on exercise type) [151], and as such, short recovery protocols may not capture the true potential for CGs to reduce limb swelling over time.

A small number of studies also reported the effect of CGs on cardiovascular measures during recovery, including heart rate, blood pressure, and blood flow. Compared to a control group, there was no change in heart rate during 30 or 80 min of recovery following maximal cycling [11,119]. Conversely, Lee and colleagues [120] reported a lower heart rate and higher stroke volume and cardiac output (as measured by Doppler ultrasound) during 60 min of recovery with CGs following submaximal cycling, with a subsequent improvement in maximal power output during a 5-min performance test. This is likely due to the choice of CGs used in this study (leg sleeves worn on top of waist-to-ankle tights), which applied a cumulative pressure of 47.4 ± 8.8 mmHg at the calf and 24.1 ± 2.4 mmHg at the thigh, which is substantially greater pressure than the average reported in recovery investigations (see Figure 2). The direct measurement of venous haemodynamics in this study affirms the ability of CGs to improve recovery by way of enhanced venous return, though future research could consider how this may influence subsequent performance longer than 5 min in duration. Additionally, Carvalho et al. [152] reported no change to mean or maximum skin temperature when compression stockings were worn for 24 h following a race pace treadmill run, though how this may change in hot environments is not known.

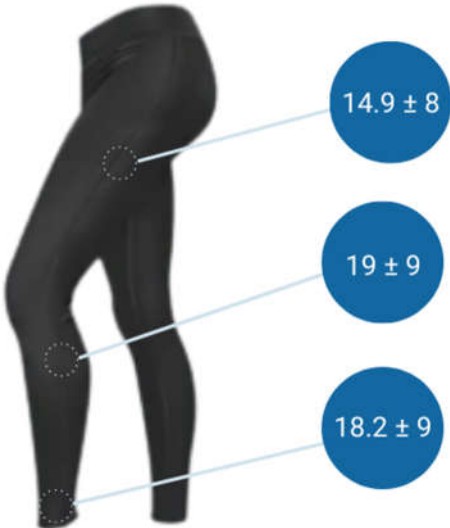

**Figure 2.** Average applied pressure (±SD) in mmHg reported at the thigh, calf and ankle for compression tights used in included 'during recovery' studies. *Note: average calculated from 16 studies which reported 'in vivo' applied pressure at the time of testing.*

Decreased heart rate, as well as decreased total peripheral vascular resistance during recovery, was also reported in Piras and Gatta [153] with the use of a custom-made full-body compression swimsuit for 90 min after a maximal 400 m swim. This has promising application for swimmers who may compete in multiple events on the same day, though as the investigation focused solely on autonomic regulation, the effect of this CG on performance and perceptual measures with regard to swimming performance is not known. Meanwhile, novice runners demonstrated an improvement in vagally-mediated heart rate variability with the use of compression tights (for 4–5 h) after daily runs for 2 weeks when compared to 'sham' tights [154]. This suggests that regular use of CGs over time can mediate some of the adverse health consequences of strenuous endurance training, e.g., autonomic nervous system imbalance [154]. Although the sample size only consisted of 10 runners, the study was strengthened by its use of a double-blind crossover design and is the only reviewed study that reported the physiological effects of chronic, rather than short-term, use of CGs for recovery.

*4.5. Perceptual Responses during Recovery*

One of the most commonly reported benefits of CGs (in the literature and, equally, in product marketing) is their potential to reduce perceived muscle soreness, such as that which occurs after strenuous or unfamiliar exercise. 35 out of the 42 reviewed studies included the measurement of at least one perceptual measure, including perceived muscle soreness, fatigue, sleep quality and pain sensations. Regarding the primary outcome of perceived muscle soreness, 20 out of 26 studies reported a decrease in muscle soreness, and six reported no change in muscle soreness with the use of CGs for recovery. The weight of the current evidence, therefore, shows that a reduction in perceived muscle soreness can be expected from the use of CGs for recovery for as little as 60 min after exercise [118,120]; and may continue to be recognised up to 96 h following exercise [144]. Further, these benefits can be recognised with the use of socks [125], tights [122], arm sleeves [144], and whole body CGs [133,139], with consideration to the muscular demands of the preceding exercise and what would be of greatest benefit to the athlete/individual. However, it should be noted that differing scales for measuring subjective muscle soreness were utilised across the research (e.g., 10 vs. 20-point visual analogue scales) and that this measure can be highly variable between subjects, contributing to low reliability [121].

In addition to reduced perceived muscle soreness, CGs used for recovery appear to reduce feelings of overall fatigue [118,124,125], improve perceived physical state and perceived recovery [134,140], increase alertness following long-haul air travel [125] and may have a *small* benefit on perceived sleep quality when worn overnight [124]. Unlike the use of CGs during performance which, in some instances, *increased* feelings of pain and tightness [27,83], there were no investigations which reported detrimental effects of CGs on perceptual responses when worn during recovery. It is, however, interesting that the two studies which used a within-subject, crossover design (use of compression on one leg only, with the other leg as the control) did not report any changes to perceived muscle soreness [142,148], lending credence to the 'placebo effect' of wearing CGs. Although, given that the garments do not appear to cause any harm and may improve the subjective perception of how the body feels (e.g., up to 34% less soreness compared to control, 24 h following an exhaustive sprint and plyometrics protocol) [121], it is clear that this 'placebo effect' has a meaningful benefit for athletes/individuals and should not quickly be dismissed.

*4.6. Combined Design (during Exercise and Recovery) Findings*

Fifteen of the included studies used a combined research design to examine the effects of CGs used for both exercise and recovery from exercise. With the exception of Bieuzen et al. [155], the same type of CG was used for both performance and recovery, with compression socks being the predominant type of garment used. In one of the earliest studies in this area, the use of compression socks during maximal cycling and 30 min

recovery resulted in lower mean lactate compared to control (no socks) and compared to wearing CGs during cycling only [50]. Without any evidence of altered lactate oxidation, the authors hypothesised this might be due to reduced diffusion of lactate from the muscular bed rather than reduced net lactate release [50]. Indeed, Rimaud et al. [80] affirmed these findings with a similarly designed study, suggesting that muscle glyconeogenesis may be favoured when lactate is retained in previously active muscle. Further studies concerning the use of CGs during both endurance performance *and* recovery have been sparse; only Ménétrier et al. [156] observed the influence of calf sleeves during and following a running TTE, showing a significant increase in oxygen saturation in the calf during 30 min of recovery.

Similar to previous discussions, the garments had minimal effects on exercise performance, including mean sprint time during repeated 20m sprints [157,158]; total distance during a 30-min intermittent repeat sprint protocol [159]; $VO_2$ max attained during an incremental cycling test [80]; or treadmill running time trial performance [156,160]. However, the use of compression tights during both a cycling task and recovery from cycling resulted in a ~6% improvement in subsequent TT performance over 8 km [161]. As well, Duffield and Portus [159] indicated no change to sprint performance (10, 20 m speed) but a *moderate* improvement in the throwing accuracy of club-level cricket players wearing a full-body Under Armour brand compression garment for performance and recovery (in comparison to both Adidas and Skins brand full-body garments). Additionally, all full-body garments resulted in a *large* reduction in CK after 24 h of wear, though garment pressure and perceived comfort during this time were not reported [159].

Overall, the use of CGs for both exercise and recovery tended to result in a decrease in perceived muscle soreness or limb pain following endurance, simulated team sport, or sprinting activities [158,159,162,163] but not following fatiguing eccentric and plyometric exercise [164]. For example, the use of compression tights during and following a fatiguing sprint protocol was associated with a *moderate* reduction in perceived muscle soreness two hours following exercise, and a significant, *large* reduction 24 h following exercise [158]. However, RPE during exercise or at the completion of exercise was unchanged in the studies that reported it [156,158,160,165].

### 4.7. Garment Considerations

A variety of styles of CGs were used across the research, though socks/stockings were the most popular choice of CG for 'during exercise' research (34% of studies), and waist-to-ankle tights were predominant for 'during recovery' research (42% of studies) (Figure 3). Clearly, the choice of garment style is determined by the choice of exercise modality being conducted, and it is evident that the current research is skewed towards lower body-dominant activities. However, this may also be partly explained by the lack of reported benefits for the use of CGs during or following upper body-dominant activities (e.g., indoor climbing) [166] and the relative lack of pressure exerted by such garments when compared to lower body garments. Custom-fitted CGs were only used in two studies for recovery [121,150] and tended to exert a higher level of applied pressure than standard-sized garments, possibly improving venous return. This may be a consideration for athletes seeking appropriately fitted CGs to optimize recovery [121].

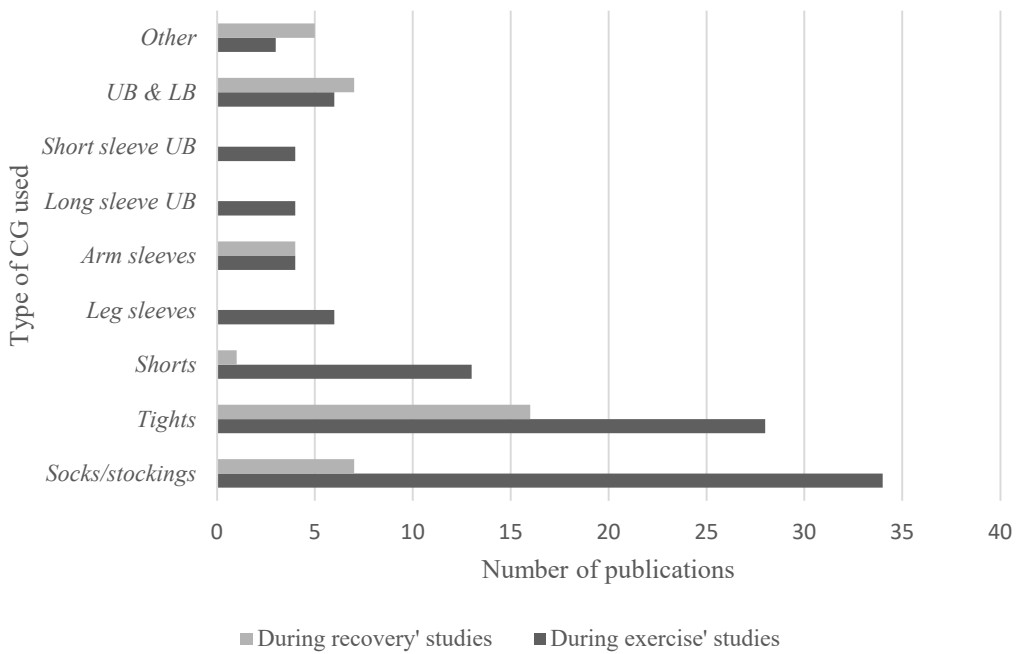

**Figure 3.** Types of compression garments used in the publications included in review. 'UB' = upper body, 'LB' = lower body, 'CG' = compression garment.

Regarding garment pressure, 40% provided an actual measurement of the chosen CGs from the time of testing (in mmHg); 25% of the reviewed studies did not report any measure of applied pressure for the CGs used, while 35% provided only manufacturer-reported estimates of applied pressure. Collecting data on the in vivo pressure exerted by the garments on the participants at the time of testing is important for subsequent research and contributes to a greater understanding of how these applied pressures may vary by participant anthropometry, as there is substantial variation in the pressures applied from the same garment based on differences in thigh and calf circumference [167]. Given that interface pressure also varies depending on posture [168], it is important that researchers state the exact details of how these pressure measures were taken, with a preference for pressure when standing to be the baseline measure [5]. It may also be useful to report the pre-exercise and post-exercise (or post-recovery) applied pressure of the CGs used, as there may be a decay of pressure during wear due to the elasticity of the fabrics used [169]. Additionally, reporting basic properties of the garment (e.g., fabric composition) is a useful detail to contribute to the literature, particularly as compression products continue to proliferate in the market.

## 5. Conclusions

The results of this systematic review can be summarised as follows:

### 5.1. During Exercise

- CGs have little to no benefit on measures of running performance (e.g., race finishing time, time to fatigue) and only small benefits to cycling performance (e.g., cycling economy, mean power during TTs).
- Countermovement jump height and power may be improved with the use of CGs (dependent on garment size and composition), but sprint performance and performance in strength and muscular endurance tasks tend to remain unchanged.
- Small improvements to sport-specific outcome variables such as T-test agility (soccer) and pitching accuracy (baseball) may be observed with the use of CGs, though research is sparse.

- While most muscle proteins/metabolites are unchanged with the use of CGs during exercise, measures of blood lactate tend to decrease compared to a control, though this is not frequently associated with significant benefits to performance.
- CGs may be positively associated with changes to soft tissue movement and muscle activation during running and sprinting activities.
- CGs tend to improve perceptions of muscle soreness and fatigue when worn during exercise.

*5.2. During Recovery*

- CGs for recovery are generally associated with positive changes to subsequent endurance exercise performance (e.g., treadmill running and cycling time trials), but not sprinting or jumping performance.
- CGs tend to improve the recuperation of maximal voluntary isometric contraction force after fatiguing resistance exercise, though research using heavy strength and power training protocols is lacking.
- Blood/plasma concentrations of lactate dehydrogenase may be reduced after recovery with CGs, but positive changes to creatine kinase, plasma IL-6, tumour necrosis factor alpha, and C-reactive protein are unlikely.
- Limb swelling following fatiguing exercise may be mitigated with the use of CGs for recovery, indicating a positive improvement in exercise-induced muscle damage.
- CGs consistently decrease measures of perceived muscle soreness following fatiguing exercise and may improve other subjective feelings such as fatigue and alertness.

*5.3. Directions for Future Research*

While there has been a substantial increase in the breadth and depth of research on CGs in the past decade, this systematic review highlights a number of key areas that could benefit from further, high-quality research (Figure 4). Firstly, the mechanism of reduced soft tissue movement with CGs during exercise has not been well elucidated, particularly as it pertains to modalities of exercise other than running. Secondly, the potential effects of long-term use of CGs on endurance and strength training adaptations have not previously been reported and may be of interest for those frequently utilising CGs as a recovery tool after training. Although previous investigations have used CGs for up to 24 h a day for 5 days [13], it is not well understood how the applied pressure of CGs may change over time and with prolonged use and frequent garment washing and drying, as these details have not been reported. Finally, the limited scope of research investigating the effects of the same CGs used for both exercise and recovery indicates further research in this area is warranted.

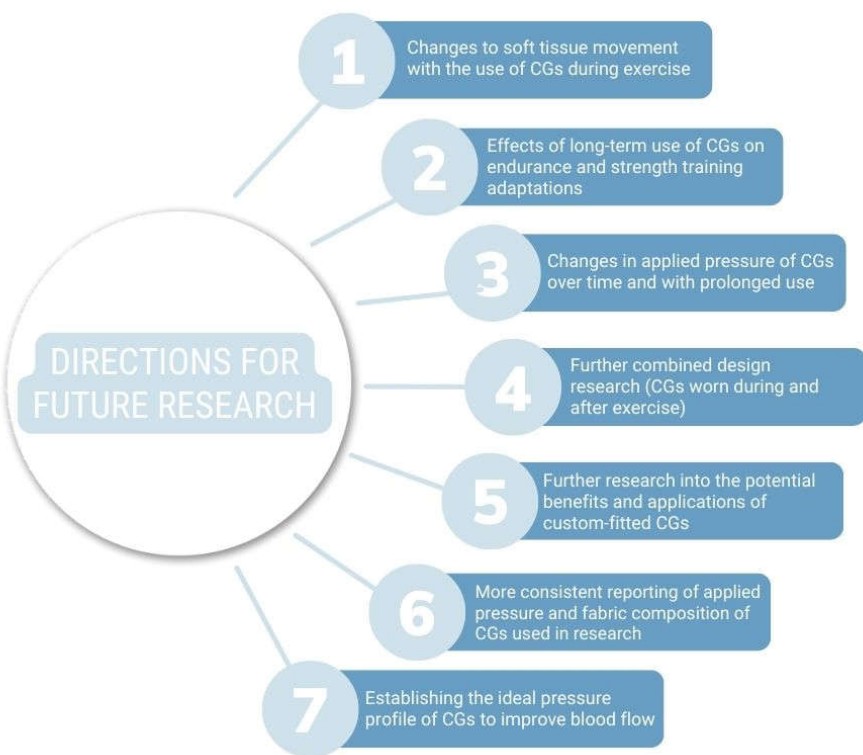

**Figure 4.** Suggested directions for future research based on the findings of the systematic review.

## 6. Perspective

Compression garments are a popular tool that may act on physiological, physical, neuromuscular, biomechanical and/or perceptual domains both during sport and exercise and for recovery from exercise, with varying levels of efficacy. Previous reviews have summarised the effects of the garments on sub-categories including running and high-intensity exercise [16,17], while a recent scoping view provided a broad summary of existing literature without individual summaries of included studies [15]. Therefore, the purpose of this review was to provide a comprehensive summary of the existing research on CGs across both exercise and recovery domains, with consideration to exercise type and all reported outcome variables. This may provide a useful point of reference for practitioners and researchers interested in the potential influence of CGs on specific outcome variables or exercise types, and also provides a guide for where future research in the field may be most valuable.

**Supplementary Materials:** The following supporting information can be downloaded at: www.mdpi.com/xxx/s1, Table S1: During exercise studies; Table S2: During recovery studies; Table S3: Combined design (during exercise & recovery) studies. References [170–187] are cited in the Supplementary Materials.

**Author Contributions:** Conceptualization, methodology, formal analysis, writing, visualization—original draft preparation, A.J.L.; methodology, supervision, writing—review and editing, M.W.D.; supervision, writing—review and editing, L.P.J. All authors have read and agreed to the published version of the manuscript.

**Funding:** This research did not receive any specific grant from funding agencies in the public, commercial, or not-for-profit sectors.

**Institutional Review Board Statement:** Not applicable.

**Informed Consent Statement:** Not applicable.

**Data Availability Statement:** Not applicable.

**Conflicts of Interest:** No potential competing interest was reported by the authors.

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
