# Peer review of "Tight Margins: Compression Garment Use during Exercise and Recovery—A Systematic Review"

_textiles, doi:10.3390/textiles2030022_

Round 1

Reviewer 1 Report

Dear Author(s),

The manuscript Textiles 2022, 2 and titled “Compression garment use during exercise and recovery: a systematic review” was reviewed. It is a regular review paper with a different review approach.

The research aim is to explain the effect of compression garment during exercise and after recovery. 

Paper is well written and it was quite explained for reader stand points. Findings from the research were acceptably discussed considering the literature, and conclusion was consistent with generated data. 

It was evaluated that it is a regular paper and finding could attract readers. The manuscript can be accepted after some revisions.

1. Please provide garment properties and possible effect of garment properties to human physiology during exercise and after recovery with possible reasoning.     

2. Please provide some information on functional garment for compression garment applications.   

3. If possible, please explain the main recovery or fatigue mode with regard to garment properties considering fiber or filament bundles. 

My best regards,

Reviewer 2 Report

The paper presents a comprehensive review with regards of compression garment use during exercise and recovery.

The authors analyzed an impressive number of literature references and synthesized an impressive amount of information.

The main issue of the paper, in my opinion is the way the information is presented to the readers. The paper includes only one figure, while all data are presented as text. Consequently, it is very hard to the reader to comprehend and follow the data offered in the paper.

Thus, I recommend the authors to introduce, wherever possible, graphics, diagrams, figures, which will make the paper much more understandable for the readers.

Reviewer 3 Report

This is a comprehensive systematic review on compression garments (CGs) for sports and exercise studies. The detailed analysis of studies presented in Tables 1-3, classified as exercise, recovery and combined studies, were commendable.

Discussions were constructive, relevant, and insightful and would serve as a reference document for compression garments.

It is interesting to note that CGs offer more benefits for the recovery of muscle soreness than during a performance.

It is also worth referring to garment considerations such as selecting the correct fit of CGs and understanding the garment sizing, which affects the applied or perceived pressure.

As specified in section 5.3, long-term usage of CGS has not been reported, and most studies monitored immediate benefits during and after activity. The authors could also highlight whether previous research reported on the assessment of applied pressure of CGs over prolonged usage and how it might vary following consumer wash or repeated wear.

Overall, an exciting and comprehensive review of compression garments worthy of publication. 

Round 2

Reviewer 2 Report

The authors tried to deal with the problem reported by the reviewer by adding graphic content to the paper. Thus, in my opinion, the paper is now much easier to follow (its scientific level was appropriate from the beginning) and I recommend that it be published.